# Synthesis of Biobased Hydroxyl-Terminated Oligomers by Metathesis Degradation of Industrial Rubbers SBS and PB: Tailor-Made Unsaturated Diols and Polyols

**DOI:** 10.3390/polym14224973

**Published:** 2022-11-17

**Authors:** Manuel Burelo, Selena Gutiérrez, Cecilia D. Treviño-Quintanilla, Jorge A. Cruz-Morales, Araceli Martínez, Salvador López-Morales

**Affiliations:** 1Institute of Advanced Materials for Sustainable Manufacturing, Tecnologico de Monterrey, Queretaro 76146, Mexico; 2Facultad de Química, Universidad Nacional Autónoma de México, Ciudad Universitaria, Coyoacán, Ciudad de México 04510, Mexico; 3Escuela Nacional de Estudios Superiores, Unidad Morelia, Universidad Nacional Autónoma de México, Antigua Carretera a Pátzcuaro No. 8701, Col. Ex. Hacienda de San José de la Huerta, Morelia 58190, Michoacán, Mexico; 4Instituto de Investigaciones en Materiales, Universidad Nacional Autónoma de México, Ciudad Universitaria, Coyoacán, Ciudad de México 04510, Mexico

**Keywords:** poly(styrene-butadiene-styrene) (SBS), tailor-made oligomers, hydroxyl-terminated polybutadiene, unsaturated polyol, unsaturated diol, metathesis degradation, catalyst optimization

## Abstract

Biobased hydroxyl-terminated polybutadiene (HTPB) was successfully synthesized in a one-pot reaction via metathesis degradation of industrial rubbers. Thus, polybutadiene (PB) and poly(styrene-butadiene-styrene) (SBS) were degraded via metathesis with high yields (>94%), using the fatty alcohol 10-undecen-1-ol as a chain transfer agent (CTA) and the second-generation Grubbs–Hoveyda catalyst. The identification of the hydroxyl groups (-OH) and the formation of biobased HTPB were verified by FT-IR and NMR. Likewise, the molecular weight and properties of the HTPB were controlled by changing the molar ratio of rubber to CTA ([C=C]/CTA) from 1:1 to 100:1, considering a constant molar ratio of the catalyst ([C=C]/Ru = 500:1). The number average molecular weight (*M_n_*) ranged between 583 and 6580 g/mol and the decomposition temperatures between 134 and 220 °C. Moreover, the catalyst optimization study showed that at catalyst loadings as low as [C=C]/Ru = 5000:1, the theoretical molecular weight is in good agreement with the experimental molecular weight and the expected diols and polyols are formed. At higher ratios than those, the difference between theoretical and experimental molecular weight is wide, and there is no control over HTPB. Therefore, the rubber/CTA molar ratio and the amount of catalyst play an important role in PB degradation and HTPB synthesis. Biobased HTPB can be used to synthesize engineering design polymers, intermediates, fine chemicals, and in the polyurethane industry, and contribute to the development of environmentally friendly raw materials.

## 1. Introduction

Polybutadiene (PB) and poly(styrene-butadiene-styrene) (SBS) are industrial rubbers with a diverse field of applications, such as soles of shoes, sealing rings, gaskets, damping materials, insulation materials, antivibration bushes, automotive parts, and the tire industry, among others, where elastomeric properties and durability are essential [1,2]. SBS is a block copolymer whose main chain is constituted by the segments polystyrene-polybutadiene-polystyrene, and butadiene is found in a greater proportion than styrene [2,3]. Industrial PB is a homopolymer in *cis*-1,4-configuration, mainly.

The tire industry is the most important market for such synthetic rubber as SBS, SBR, PB, butyl rubber (IIR) and polyisoprene (IR). Styrene butadiene rubber is the most important grade at the moment, with around 5.1 million tons consumed in 2020. In 2020, 14.4 million tons of synthetic rubber were produced. The global demand for tires will reach 3.2 billion units in 2022. Furthermore, it is reported that a total of about 223 million replacement tires were shipped to the USA in 2019 [4,5]. It is important to note that, once their use is over or they are at the end of their life, these materials are disposed of in landfills or end up as microplastics (MPs) in the oceans. For example, the deterioration of the tires due to their abandonment or rolling produces small fragments that travel through the air, land, and water, becoming part of the MPs. Moreover, rubber-based materials are difficult to degrade naturally in landfills due to their chemical composition, high molecular weight, cross-linking, and additives, among others. Thus, large quantities of these materials accumulate in the environment, causing its deterioration and contamination [6,7].

In addressing this challenge, several waste-management methods have been investigated and tested in the last few decades, including thermal (pyrolysis), mechanical (mechanical or cryogenic trituration), physical (radiation, ultrasonic radiation, or microwave), chemical (catalyst systems, ozonolysis, free radicals, or oxidation), and biological (fungi, bacterial strains, or microbial communities) processes [6,7,8,9,10,11,12,13,14]. According to reports, these methods show some disadvantages. For instance, they require high temperatures and/or pressure, the use of large quantities of solvents, involve the generation of different types of toxic gases or by-products and several steps in the reactions, low yields, wide variety of products, low conversions, and most of all, there is no control over the degradation process and thus over the structure and molecular weight of the oligomers. These facts limit the use or application of the products obtained and prevent a waste-rubber recycling path.

On the contrary, the depolymerization process via metathesis has shown that both natural and industrial rubbers can be degraded under mild conditions, at low temperatures and atmospheric pressure, in solvent-free conditions, or allowing the use of essential oils as green solvents [15,16,17,18,19]. Metathesis allows the design of oligomers by carefully selecting the olefins used as chain transfer agents (CTA) and the polymer/CTA molar ratio. For this reason, oligomers via metathesis have been called tailor-made oligomers or telechelic oligomers due to the high control over the chemical structure, molecular weight, and functional group number per chain (*Fn* close to 2). Telechelic compounds, oligomers, polymers, biobased products, and raw materials with different functional groups have been obtained via metathesis degradation reaction, including hydroxyl-terminated compounds, which represent a challenging task [15,16,17,18,19].

Metathesis reactions are limited to compounds with double bonds [C=C] in their structure and catalysts based on transition metals such as Ru, Mo, and W. Highly active catalysts based on Ru and high concentrations of these are required to achieve the successful synthesis of hydroxy-terminated compounds [19,20]. In the last few years, the authors have reported that several factors, such as the CTA, selected catalyst, reactant concentration, solvent, and olefin ring-strain can affect the metathesis degradation reaction and synthesis of hydroxy-terminated compounds [20,21,22]. In this sense, unprotected alcohols used as CTAs are not fully compatible with the metathesis catalysts.

Hydroxyl-terminated polybutadiene oligomers (HTPB) can be used as diols and polyols. Diols are employed in the elaboration of adhesives, coating, sealants, intermediate products, fine chemicals, and polymeric binders for both propellants and explosives, and as chain-extender oligomers in polycondensation reactions for the synthesis of copolymers, polyether polyols, and polyurethanes (PU), where they constitute the rigid segment [23,24,25,26]. Meanwhile, polyols are used in the synthesis of polymers and copolymers by polycondensation reactions. In polyurethanes, the polyol constitutes the flexible segments in the main chain and determines the type of PU that can be obtained and its applications [27,28,29,30].

On the other hand, most of the raw materials employed for polymer production (oligomers, diols, polyols, and isocyanates) are derived from nonrenewable petroleum resources. Currently, the chemical industry and scientific community are focused mainly on replacing fossil raw materials with environmentally friendly alternatives and on developing materials that, are suitable for recycling or biodegradation at the end of the product’s life [31,32]. For instance, terpenes, vegetable oils, and carbohydrates have been used as feedstock to manufacture various sustainable raw materials and products. The fatty alcohol 10-undecen-1-ol can be obtained from castor oil (*Ricinus communis*), and it has been converted into diols and polyols [33,34,35,36]. In that sense, we report the synthesis of biobased hydroxyl-terminated oligomers (HTPB) via metathesis degradation from polybutadiene and poly(styrene-butadiene-styrene) in a one-pot reaction using the fatty alcohol 10-undecen-1-ol as CTA and the second-generation Hoveyda–Grubbs catalyst Ru, as well as the catalyst optimization for this process.

## 2. Materials and Methods

### 2.1. Materials and Reagents

*Cis*-1,4-polybutadiene (PB) (M_n_ = 9.10 × 10^5^ g/mol, PDI = 2.20) was obtained from Mexico (Compañía Hulera, Ciudad de México, Mexico). Poly(styrene-butadiene-styrene) (SBS) (styrene, 30 wt.%) (M_n_ = 1.70 × 10^5^, PDI = 1.50), 10-undecen-1-ol (98%) (CTA), chlorobenzene anhydrous, methanol, and the catalyst [1,3-bis-(2,4,6-trimethylphenyl)-2-imidazolidinylidene]dichloro(o-isopropoxyphenylmethylene) ruthenium (second generation Hoveyda-Grubbs) (Ru) were purchased from Sigma-Aldrich, Inc. (St. Louis, MO, USA), and used as received.

### 2.2. Synthesis of Biobased Hydroxyl-Terminated Oligomers (Biobased HTPB) via Metathesis Degradation from SBS or PB

The synthesis of biobased HTPB was carried out via metathesis degradation reaction from industrial rubber (SBS or PB) using the fatty alcohol 10-undecen-1-ol as chain transfer agent (CTA), chlorobenzene as solvent, and second-generation Hoveyda–Grubbs catalyst Ru (Figure 1). All reactions were performed under a nitrogen atmosphere using standard Schlenk line techniques in a 100 mL Schlenk tube containing a Spinplus stir bar from Sigma-Aldrich.

First, the rubber SBS or PB (3.00 g, 55.50 mmol) and CTA (9.44 g, 55.50 mmol for molar ratio rubber/CTA = 1:1) were placed in the Schlenk tube charging with chlorobenzene anhydrous as solvent (10 wt.%), then the catalyst Ru (0.13 g, 0.2220 mmol for C=C/Ru = 500) was added. The temperature was controlled using an oil bath maintained at 40 °C for 12 h with continuous stirring.

The molar ratios of carbon-carbon double bonds of SBS or PB to CTA used were [SBS]/[CTA] or [PB]/[CTA] = 1:1, 10:1, 20:1, 50:1, and 100:1 (9.44, 0.94, 0.47, 0.18 and 0.094 g, respectively) in order to control the molecular weights and for the catalyst [C=C]/[Ru] = 500. During the reaction process, ethylene is generated as a by-product, which means that is possible to displace the equilibrium process toward a specific product. Thus, in order to favor the formation of HTPB, ethylene gas was removed using a vacuum line [16,19]. After terminating the reaction by the addition of a small amount of ethyl vinyl ether (0.30 mL, 3.00 mmol), the product was precipitated three times in methanol and purified. In the case of SBS, the polystyrene microblocks (gray solid) were separated by precipitation in a methanol solution and then by decantation using a separation funnel. The product was isolated and dried under vacuum (Thermo Scientific vacuum oven, Waltham, MA, USA) at 60 °C for 8 h. Finally, the biobased HTPB was weighed and the yield calculated by the gravimetric method. The HTPBs (oligomers or polyol) were both sandy brown liquids with different molecular weights with a yield between 94% and 98%. The data are summarized in Table 1.

The structure, molecular weights and thermal properties of the biobased HTPBs were characterized by infrared spectroscopy (FT-IR), nuclear magnetic resonance (NMR), gel-permeation chromatography (GPC) and thermogravimetric analysis (TGA and DTG). The reactions were carried out in triplicate and analyzed (*n* = 3) for each reaction using means ± standard deviation for GPG and the yields.

### 2.3. Catalyst Optimization for the Synthesis of Biobased HTPB Oligomers by Metathesis Degradation

Catalyst optimization for biobased HTPB synthesis was performed via metathesis degradation reaction of PB (3.00 g, 55.50 mmol) using the fatty alcohol 10-undecen-1-ol as CTA (9.44, 0.94 and 0.094 g) and second–generation Hoveyda–Grubbs catalyst (Ru) (Figure 1). In all reactions, similar conditions to those described in Section 2.2 were maintained, except for the molar ratio of the catalyst, which ranged between 500–10,000. The molar ratios [C=C]/[Ru] = 500:1, 1000:1, 2000:1, 5000:1 and 10,000:1 (0.1222, 0.0611, 0.0305, 0.0122 and 0.0061 mmol of Ru, respectively) were tested in the experiments. Furthermore, the molar ratio of PB to CTA was kept constant at [PB]/[CTA] = 10:1 (0.94 g, 5.55 mmol) in the PB2, and PB4–PB7 reactions (Table 2). Thus, considering the number of repeating units of PB (m = 10), and the terminal groups from the CTA (10-undecen-1-ol), the theoretical molecular weight of oligomers is 852 g/mol (Equation (1)). After terminating the reaction, the product was precipitated three times in methanol and purified. The products were isolated and dried under a vacuum (Thermo Scientific vacuum oven, Waltham, MA, USA) at 60 °C for 8 h. Finally, the biobased HTPB was weighed and the yield calculated by the gravimetric method. The HTPBs (oligomers or polyol) were both sandy brown liquids with different molecular weights with a yield between 94% and 98%. The data are summarized in Table 2. The structure, molecular weights and thermal properties of the biobased HTPBs were characterized by NMR, GPC, and TGA. The reactions were carried out in triplicate and analyzed (n = 3) for each reaction using means ± standard deviation for GPG and the yields.

### 2.4. Measurements and Characterizations

Infrared spectroscopy (FT-IR) measurements of all samples were analyzed using a Perkin Elmer Frontier MIR FT-IR spectrometer (Waltham, MA, USA) fitted with a Frontier Universal Diamond/ZnSe ATR with a single reflection top plate and pressure arm. The spectra were recorded in the region of 4000–500 cm^−1^. Data are presented as the frequency of absorption (cm^−1^).

Nuclear magnetic resonance (^1^H-NMR) spectra were recorded using a Bruker spectrometer at 400 MHz (Billerica, MA, USA) and deuterated solvents (chloroform-d, CDCl_3_), and chemical shifts (δ) are reported in parts per million (ppm) relative to tetramethylsilane (TMS) as an internal standard.

The number and weight average molecular weight (*M_n_* and *M_w_*) and molecular weight distribution (PDI) were determined with reference to monodisperse polystyrene standards on a Waters 2695 ALLIANCE gel-permeation chromatography (GPC) instrument (Milford, MA, USA) at 30 °C with tetrahydrofuran (HPLC grade) as a solvent, a universal column, and a flow rate of 1 mL/min. At least three replicates were analyzed for each condition using means ± standard deviation for *M_n_*, *M_w_* and PDI.

Thermogravimetric analysis (TGA and DTG) was carried out on a TA Instrument Q5000 (New Castle, DE, USA) under a nitrogen atmosphere from room temperature to 600 °C with a heating rate of 10 °C/min.

## 3. Results and Discussion

### 3.1. Synthesis of Biobased Hydroxyl-Terminated Oligomers via Metathesis Degradation from Poly(styrene-butadiene-styrene)

In Section 3.1, Section 3.2, Section 3.3 and Section 3.4 the results obtained for HTPB degrading SBS rubber and their properties are described. In Section 3.5 the results obtained for the HTPB degrading PB rubber and the catalyst optimization used in the reactions are described. The synthesis of biobased hydroxyl-terminated oligomers (biobased HTPB) was carried out via metathesis degradation reaction of SBS using the fatty alcohol 10-undecen-1-ol as CTA with different molar ratios of rubber/CTA in the presence of the Ru-alkylidene catalyst (Ru) (Figure 1). It is worth noting that biobased HTPB can be classified as unsaturated diols and unsaturated polyols, both with the same structure but different molecular weights. According to the reactions shown in Table 1, the SBS was degraded in a controlled manner with high yields, in a range of 94–98%, considering that the SBS copolymer was constituted by microblocks of PB/PS in 70/30 wt.%.

### 3.2. FT-IR and NMR Analysis

FT-IR and NMR analysis confirmed the formation of HTPB. Figure 1 displays the comparative FT-IR spectra of SBS before (Figure 1a) and after their cross-metathesis degradation using the 10-undecen-1-ol as CTA (Figure 1b). In Figure 1a, the characteristic signals of SBS are observed. The absorption peak at 3025–3007 cm^−1^ was attributed to the stretching of the double bond (-CH=CH-), the absorption peaks at 2925–2851 cm^−1^ were attributed to the stretching of the aliphatic symmetric and asymmetric methylene groups (-CH_2_-), and the absorption peaks at 1650–1436 cm^−1^ were attributed to the stretching of double bonds (C=C) present in the aliphatic chain and aromatic ring in the SBS. Finally, the signals at 1000–970 and 698 cm^−1^ are associated with the aromatic ring present in the polystyrene (PS).

In contrast with Figure 1a, Figure 1b shows a broad absorption band at 3300–3250 cm^−1^, which is associated with the R-OH stretching vibration of the HTPB. The signals of aromatic rings at 1000–970 and 698 cm^−1^ disappeared, correlated with the participation of SBS in the degradation reaction via metathesis, and the formation of the expected biobased HTPB, after isolating the microblocks of the PS. In FT-IR spectra, two features indicate the degradation in polymers: changes in signal intensity and the appearance of new signals [30,37].

The formation of biobased HTPB also was confirmed by ^1^H-NMR. Figure 2 shows the comparative ^1^H-NMR spectra of SBS before (Figure 2a) and after their metathesis degradation using the 10-undecen-1-ol as CTA (Figure 2b), and polystyrene isolated and recovered from the reaction (Figure 2c). In spectrum (a), the characteristic signals of SBS are observed. The signals of the aromatic ring at 7.3–6.3 ppm (from c to e), and the signals of -CH protons and methylene protons (-CH_2_-) at 1.8–1.25 ppm (a and b) due to the aliphatic chain of styrene. The signal at 5.4 ppm (g) corresponds to the olefinic protons (-CH=CH-) of PB and the signal at 2.1–1.9 ppm (f) corresponds to methylene protons (-CH_2_-). In spectrum (b), the signals that appeared in the range of 5.70–5.55 ppm were attributed to the formation of the double bonds (HC=CH) between the PB and 10-undecen-1-ol and the signal at 3.52 ppm (j) corresponds to the methylene protons attached to the hydroxyl groups (-CH_2_-OH). This confirms the participation of the PB microblocks of the SBS in the metathesis degradation and the formation of biobased HTPB. In spectrum (c), the signals of the PS microblocks isolated and recovered from the reaction are observed. The signals of the aromatic ring, -CH protons, and methylene protons were previously described in the spectrum (a). As can be seen, the PS microblocks can be recovered from the metathesis degradation reaction of SBS, as shown in Figure 1, and could be reused as oligomers or prepolymers for new polymerization reactions. NMR analysis agrees with similar spectra reported for pure PB, SBS, and natural rubber in metathesis reactions with essential oils and fatty alcohols [19,38].

### 3.3. Molecular Weight Analysis by GPC

The degradation process of a polymer implies that the polymeric chains are cleaved, causing a decrease in its molecular weight (*M_n_ or M_w_*) and a change in the polydispersity index (PDI). Table 1 shows the molecular weights obtained by GPC for the biobased HTPB. The results showed that using different molar ratios of double bonds [C=C] of rubber [SBS] to [CTA] (from 1:1 to 50:1), the molecular weight of the oligomers can be controlled, considering as a reference the theoretical molecular weight. The decrease in *M_n_* and *M_w_* is attributed to the production of smaller chains of unsaturated diols and polyols. Thus, biobased HTPB with *M_n_* ranging from 583 to 3583 g/mol were obtained when SBS was degraded in a controlled manner (Entries SBS1-SBS4, Table 1). It is worth mentioning that the molecular weight (*M_n_*) of SBS before degradation was 170,000 g/mol.

Figure 3 shows the *M_w_* and molecular weight distribution (*MWD*) of biobased HTPB. Different *MWDs* are observed, and lower molecular weights shift to the left. The decrease in *M_w_* of SBS when this was degraded via metathesis (Entries SBS1-SBS4, Table 1) confirms the formation of oligomers. Changes and decreases in *M_n_, M_w_* and *MWD* have been reported in rubber degradation studies [19,21,39], which agrees with the changes reported in this work by GPC.

### 3.4. Thermal Properties by TGA and DTG

Values of the decomposition temperature (*T_d_*) of biobased HTPB formed by metathesis degradation reaction of SBS are summarized in Table 1. *T_d_* was studied by TGA from room temperature to 600 °C, showing different percentages for mass loss (5 and 50%). Figure 4a shows TGA thermograms for the SBS oligomers, which suffer a mass loss of 5%, at 134, 147, 152, and 158 °C, for the Entries SBS1-SBS4 (Table 1), respectively. As can be seen, an increase in *T_d_* was recorded as the molecular weight of the biobased HTPBs also increased. A similar trend was observed for *T_d_* 50%, with temperatures ranging from 400 to 429 °C.

Figure 4b depicts the DTG thermograms from biobased HTPB obtained of SBS degradation (Entries SBS1-SBS4, Table 1), showing for SBS4 a larger area and a right shift at higher temperature values (higher molecular weight oligomer) than SBS1 (lower molecular weight oligomer). Likewise, two degradation steps were observed. The peaks observed at 150–250 °C and 400–450 °C are directly related to the decomposition of hydroxyl and olefinic bonds, respectively [33,40]. Therefore, the decomposition temperature of biobased HTPB depends on the molecular weight.

As shown earlier, the metathesis degradation reactions of polyalkenamers in the presence of unsymmetrical CTAs leads to a set of oligomers. In this sense, such olefins reduce the control over the reaction. However, this problem can be circumvented using a terminal olefin as CTA, since the metathesis is a thermodynamically controlled reaction and reaches equilibrium. As a consequence, the formation of ethylene shifts the reaction towards the products [16,19,33]. In Table 1, the 10-undecen-1-ol was used as CTA in the metathesis degradation reaction of industrial rubbers, and biobased HTPB were obtained with a high yield, thanks to the fact that ethylene formed during the metathesis degradation was removed by pressure release (Figure 1). It has been reported that operating under reduced pressure ensures the removal of the volatile olefin [41]. Likewise, it was reported that when the 10-undecen-1-ol is used as CTA in self-metathesis or cross-metathesis reactions of PB or natural rubber, the expected products unsaturated diol and polyol were widely favored and the by-products (oligomers terminated by -OH on one end and =CH_2_ on the other, α,ω-vinyl terminated oligomers, and ethylene) were formed with low yields. These results are in agreement with this work and other reported work for the metathesis of 10-undecen-1-ol as CTA [19,33].

### 3.5. Catalyst Optimization for the Synthesis of Biobased HTPB via Metathesis Degradation from PB

This study synthesized biobased HTPB by metathesis degradation reaction between PB and the fatty alcohol 10-undecen-1-ol (Figure 1). The molar ratios of the double bonds [C=C] of PB to CTA used were [PB]/[CTA] = 1:1, 10:1 and 100:1 (Table 2), keeping the molar ratio constant (10:1) in the PB2 and PB4–PB7 reactions and varying the molar ratio of Ru catalyst from [C=C]/[Ru] = 500–10,000.

According to the degradation reactions (Table 2), the PB was degraded in a controlled manner with high yields of 94–98%. It is worth mentioning that the PB industrial rubber’s molecular weight (*M_n_*) before degradation was 910,000 g/mol. Thus, biobased HTPB with *M_n_* ranging from 643 to 6580 g/mol were obtained.

Table 2 shows that the molecular weights (*M_n_*) obtained by GPC for the synthesized PB2 and PB4–PB7 oligomers with molar ratio constant (10:1) ranged from 873 to 1753 g/mol. The theoretical molecular weight (*MW*) of oligomers determined by Equation (1) was 852 g/mol.
(1)$$Theoretical\;MW=MM_{MU}\times m+MM_{CTA}\;$$
$$MW=\frac{54g}{mol}\times 10+\frac{312g}{mol}=852\;g/mol\;$$
where,

*Theoretical MW* = *theoretical molecular weight (g/mol)*.

*MM_MU_* = *molecular mass of the monomeric unit, for butadiene* = 54 *g/mol*

*m* = *repetitive units*, *m* = 10. *It was fixed according to molar ratio PB/CTA*.

*MM_CTA_* = *molecular mass of CTA* = 312 *g/mol* (*at both chain ends*, Figure 1).

According to the results obtained using catalyst molar ratios [C=C]/[Ru] of 500–5000 (0.1222–0.0122 mmol of Ru), it was possible to control the molecular weight of oligomers (*M_n_*). The theoretical molecular weight is very similar to the experimental one. In contrast, when a molar ratio of [C=C]/[Ru] = 10,000, the experimental molecular weight was practically twice the theoretical (Table 2). These results confirm that the molar ratio and the catalyst amount play an important role in the degradation of PB and in control of the number average molecular weight (*M_n_*) for the synthesis of biobased hydroxyl-terminated oligomers (Figure 5).

NMR analysis was carried out for the biobased HTPB synthesized to verify that the degradation reactions can be controlled, i.e., the structure and molecular weight of the oligomers, changing the concentrations of the catalyst.

Figure 6 shows the spectra of PB2 and PB4–PB7 (Entries 2, 4–7, Table 2). The signals that appeared in the range of 5.7–5.40 ppm (signals g and h) were attributed to the formation of the double bonds (HC=CH) between the monomeric unit of butadiene and 10-undecen-1-ol, the signal at 2.1 ppm (f) corresponds to methylene protons (-CH_2_-), and the signal at 3.52 ppm (j) corresponds to the methylene protons attached to the hydroxyl groups (-CH_2_-OH). This signal (j) is essential in the spectra because—in addition to confirming the formation of oligomers—it provides a measure of control in molecular weight by functional groups. Unlike the PB7 spectrum, where a decrease in the signal (j) is observed at 3.52 ppm, this is associated with the value of repeating units of PB (*m* different from 10) and its molecular weight. Therefore, at molar ratios greater than 10,000:1, there is no control in the molecular weight of biobased HTPB.

On the other hand, the *T_d_* at 5 and 50% were obtained by TGA for the biobased HTPB obtained by metathesis degradation reaction of PB (Entries PB2, PB4-PB7, Table 2). Figure 7 shows that the temperature values and the tendency of thermograms for the biobased HTPB are similar, since they have the same structure and similar molecular weights (except PB7). Therefore, using low concentrations of Ru catalyst can be synthesized biobased HTPB with similar molecular weights and decomposition temperature (*T_d_*).

Therefore, for metathesis reactions using the industrial rubbers SBS or PB, it is possible to use low concentrations of catalyst to carry out the reaction. It is reported that for metathesis reactions of rubbers, a ratio molar from 500:1–4000:1 of catalyst is used [15,19,38]. On the other hand, when natural or polyisoprene rubber is used, the reactions need an adequate concentration of catalyst, ideally from 250:1 to 1000:1 [18,19,39]. Unlike PB or SBS, natural or polyisoprene rubber requires the use of major concentrations of catalyst. This is due to their being polymers with trisubstituted unsaturations with the presence of an alkyl group directly linked to the double bond, which can impede the coordination reaction with an active metal center of the catalyst [19,22,42].

Soon, the waste derived from SBS and PB was degraded via cross-metathesis using 10-undecen-1-ol (which can be obtained from castor oil) and other natural resources, such as CTA, optimizing and economizing the catalyst. Biobased HTPB obtained with lower and higher molecular weight (tailor-made unsaturated diols and polyols, respectively) can be used to synthesize engineering design polymers, intermediates, fine chemicals, and in the polyurethane industry and contribute to developing environmentally friendly raw materials: “reuse–reduce–recycle.”

## 4. Conclusions

Biobased HTPB (unsaturated diols or polyols by their molecular weight) were synthesized by metathesis degradation from PB and SBS using the fatty alcohol 10-undecen-1-ol as CTA and the second-generation Grubbs–Hoveyda catalyst with high yields (94–98%). The PB was degraded, as well as the PB microblocks of the SBS. Meanwhile, the polystyrene (PS) microblocks of such copolymer were isolated and recovered. Moreover, it was established that by changing the molar ratio of double bonds [C=C] of rubber (PB or SBS) to CTA in a wide range (1:1–100:1), the molecular weight and the properties could be controlled at a constant molar ratio of the catalyst ([C=C]/Ru = 500:1). The number average molecular weight (*M_n_*) obtained ranged from 583 to 6580 g/mol, and decomposition temperatures (*T*_*d* 5%_) from 134 to 220 °C.

The rubber/CTA molar ratio and the amount of catalyst ([C=C]/Ru) play an important role in PB degradation. The catalyst optimization study showed that at catalyst loadings as low as [C=C]/Ru = 5000:1, the theoretical molecular weight is in good agreement with the experimental molecular weight, and the expected diols and polyols are formed. The difference between theoretical and experimental molecular weights is wide at higher ratios than those, and there is no control over biobased HTPB.

Therefore, the chosen CTA and catalyst synthesized tailor-made biobased HTPBs by metathesis degradation of industrial rubbers in a one-pot reaction. The second-generation Grubbs–Hoveyda catalyst has an internal coordination bond that makes it tolerant to air, moisture, and most functional groups (-OH).

Biobased HTPB can be used to synthesize engineering design polymers, intermediates, fine chemicals and in the polyurethane industry, and contribute to the development of environmentally friendly raw materials: “reuse–reduce–recycle.”

## Data Availability

Not applicable.

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
