# Peer review of "Synthesis of Biobased Hydroxyl-Terminated Oligomers by Metathesis Degradation of Industrial Rubbers SBS and PB: Tailor-Made Unsaturated Diols and Polyols"

_polymers, 2022, doi:10.3390/polym14224973_

Round 1
Reviewer 1 Report
This paper investigates the synthesis of biobased hydroxyl-terminated oligomers by metathesis degradation of industrial rubbers SBS and PB, and the catalyst optimization. This research is very interesting and can make full use of industrial rubbers, and can be contribute to the development of environmentally friendly raw materials. This paper can be accepted after modification.
(1) The details of the test, such as the pressure of the test vessel during the reaction process and the weight of SBS, PB, and CTA used in this study. The details of the test design are not prominent, and the repeatability of other scholars is low.
(2) Section 2.4, the number of test groups in the process of measurements and characterizations needs to be supplemented.
(3) The results and discussion section should indicate whether the data is average? Is the data discrete? Error between test groups?
(4) In the results and discussion section, the author should consider the reliability of the data results and how to ensure the reliability of the data results.
(5) The error line of data should be added in Figure 5 due to the reliability of the test results of a single group of data is low.
Author Response
We thank reviewer #1 for the revision of our manuscript and comments. Please find below our detailed response to each comment.
1. The details of the test, such as the pressure of the test vessel during the reaction process and the weight of SBS, PB, and CTA used in this study. The details of the test design are not prominent, and the repeatability of other scholars is low.
Answer. The synthesis of biobased HTPB has been described in detail in section 2.2, in order to enhance the reproducibility.
The reaction procedure was explained step by step, and details about the pressure and the weights of the reagents used, as well as the weights of the products obtained as reaction yields calculated by gravimetry, were included in the revised version of the manuscript (page 3, line 123).
The degradation reactions via metathesis occur between the C=C double bonds of the rubber and the ATC. So all calculations are in molar ratios. The quantity of CTA and the catalyst (Ru) vary in the reactions according to the molar ratios of Rubber/CTA and C=C (rubber+CTA)/Catalyst. They are used and presented in relation to the amount of rubber used (3.0 g) and calculated by stoichiometry. These molar ratios are used for reactions via metathesis, some reports by other authors:
1. Jiang, B.; Wei, T.; Zou, T.T.; Rempel, G.L.; Pan, Q.M. A Novel Approval for Degradation of Polybutadiene and Synthesis of Diene-Based Telechelic Oligomers via Olefin Cross Metathesis. Macromol React Eng 2015, 9, 480–489. https://doi.org/10.1002/mren.201500004
2. Martínez, A.; Zuniga-Villarreal, N.; Gutierrez, S.; A. Tlenkopatchev, M. New Ru-Vinylidene Catalysts in the Cross-Metathesis of Natural Rubber and Poly(Styrene-Co-Butadiene) with Essential Oils. Curr Org Synth 2016, 13, 876–882.
http://dx.doi.org/10.2174/1570179413666151218203008
3. Bielawski, C.W.; Benitez, D.; Morita, T.; Grubbs, R.H. Synthesis of End-Functionalized Poly ( Norbornene ) s via Ring-Opening Metathesis Polymerization. Macromolecules 2001, 34, 8610–8618. https://doi.org/10.1021/ma010878q
2. Section 2.4, the number of test groups in the process of measurements and characterizations needs to be supplemented.
Answer. All reactions were carried out in triplicate, as well as their analysis. FT-IR and NMR spectra were used to analyze the chemical structure, and representative spectra were selected for include in the manuscript. For the thermal analysis, we proceed in a similar manner (Tables 1 and 2). On the other hand, the Mn, Mw and PDI were calculated by GPC, and the reaction yields by gravimetry. For these data, at least three replicates were analyzed for each condition, n=3; standard deviation and percentage of error were calculated. The information was included in the revised version of the manuscript, section 2.2 (page 4, line 154), section 2.4 (page 4, lines 194-195), and summarized data in Tables 1 and 2 (pages 5 and 10).
3. The results and discussion section should indicate whether the data is average? Is the data discrete? Error between test groups?
Answer. This comment was clarified in the previous answer.
4. In the results and discussion section, the author should consider the reliability of the data results and how to ensure the reliability of the data results.
Answer. All reactions were carried out in triplicate, and analysis (n= 3) for each reaction. It was considered n = 3, mean ± standard deviation for Mn, Mw, PDI, and the reaction yields.
Moreover, all the results obtained by FT-IR, NMR, GPC and thermal properties are discussed in the results section and compared with other works reported for rubbers degradation and the techniques used, in which similar results are reported.
Further, for the results obtained, the equipment used is certified, and the techniques in this research (FT-IR, NMR, GPC and thermal properties) have been used and reported for our research group for the degradation of rubbers and polyurethanes:
1. Burelo, M.; Gaytán, I.; Loza-Tavera, H.; Cruz-Morales, J.A.; Zárate-Saldaña, D.; Cruz-Gómez, M.J.; Gutiérrez, S. Synthesis, Characterization and Biodegradation Studies of Polyurethanes: Effect of Unsaturation on Biodegradability. Chemosphere 2022, 307, 136136. https://doi.org/10.1016/j.chemosphere.2022.136136
2. Burelo, M.; Martínez, A.; Cruz-Morales, J.A.; Tlenkopatchev, M.A.; Gutiérrez, S. Metathesis Reaction from Bio-Based Resources: Synthesis of Diols and Macrodiols Using Fatty Alcohols, β-Citronellol and Natural Rubber. Polym Degrad Stab 2019, 166, 202–212. https://doi.org/10.1016/j.polymdegradstab.2019.05.021
3. Martínez, A.; Tlenkopatchev, M.A.; Gutiérrez, S.; Burelo, M.; Vargas, J.; Jiménez-Regalado, E. Synthesis of Unsaturated Esters by Cross-Metathesis of Terpenes and Natural Rubber Using Ru-Alkylidene Catalysts. Curr Org Chem 2019, 23, 1356–1364. https://doi.org/10.2174/1385272823666190723125427
5. The error line of data should be added in Figure 5 due to the reliability of the test results of a single group of data is low.
Answer. We appreciate the comments. Suggested information was included in the revised version of the manuscript in Table 2 (page 10), and Figure 5 has been corrected. For Figure 5, the data of molar ratio (X-axis) and the number of moles (Y1-axis) are constant and calculated by stoichiometry, were added the data and the error line for the molecular weight Mn (Y2-axis).

Reviewer 2 Report
A very interesting article on the issues of rubber waste management, including car tires.
The authors described one of the methods of car tire disposal using fatty alcohol for the degradation of PB and SBS. The influence of the catalyst on the course of the degradation of industrial rubbers was demonstrated.
HTPBs obtained as a result of biochemical treatments can be used for the synthesis of polymers for engineering design, intermediates, fine chemicals and in the polyurethane industry, and contributes to the development of environmentally friendly raw materials: "Reuse-Reduce-Recycle".
Authors demonstrated proficiency in using modern analytical tools and presented the results of their work in an original way.
General remarks on the manuscript:
1. During the review of industrial rubber degradation methods, the authors omitted one of the innovative methods described in the article by Pietras, Olejnik, Śliżewska, Sielski, Sobiecka: "The process of natural and styrene-butadiene rubbers biodegradation by Lactobacillus plantar", Appl. Sci. (2022), 12, 5148, https://doi.org/10.3390/app12105148
2. The included analyzes lacked calculations concerning the energy balance during the reduction of rubbers. Such data would significantly emphasize the environmental aspect of the research.
Author Response
Reviewer #2:
We thank reviewer #2 for the revision of our manuscript and comments. Please find below our detailed response to each comment.
1. During the review of industrial rubber degradation methods, the authors omitted one of the innovative methods described in the article by Pietras, Olejnik, Śliżewska, Sielski, Sobiecka: "The process of natural and styrene-butadiene rubbers biodegradation by Lactobacillus plantar", Appl. Sci. (2022), 12, 5148, https://doi.org/10.3390/app12105148
Answer. Thank you for your comment. We included the biodegradation method of SBS rubber using bacterial strain and the suggested reference in the revised version of the manuscript (page 2, lines 62-63, Ref. [14]).
2. The included analyzes lacked calculations concerning the energy balance during the reduction of rubbers. Such data would significantly emphasize the environmental aspect of the research.
Answer. We appreciate the comments. The reaction time and temperature were set in the degradation processes at 12h and 40 °C to reach equilibrium in the reactions and the complete conversion of the reactants and products. An energy balance during the degradation process was not designed for this study, focusing on the mass losses and molecular weight distribution calculated by GPC before and after the degradations via metathesis, on the thermal properties and the influence of the catalyst during the degradation of industrial rubbers. However, in the middle term, it is planned to study different times and temperatures during the degradation process, optimize the times to use the least energy and be able to carry out an energy balance in the degradation process, enthalpies, thermal energy study and analysis on the carbon footprint during the rubber degradation process.
Kind regards,
Dr. Manuel Burelo,

Round 2
Reviewer 1 Report
Authors have revised the manuscript very carefully, my comments are well addressed, there is no further comments from my side. Good work.